# Data-Driven Insights through Industrial Retrofitting: An Anonymized Dataset with Machine Learning Use Cases

**DOI:** 10.3390/s23136078

**Published:** 2023-07-01

**Authors:** Daniele Atzeni, Reshawn Ramjattan, Roberto Figliè, Giacomo Baldi, Daniele Mazzei

**Affiliations:** 1Department of Computer Science, University of Pisa, 56126 Pisa, Italy; reshawn.ramjattan@phd.unipi.it (R.R.); roberto.figlie@phd.unipi.it (R.F.);; 2Zerynth, 56124 Pisa, Italy; g.baldi@zerynth.com

**Keywords:** industry 4.0, retrofit, machine learning, benchmark dataset, industrial IoT

## Abstract

Small and medium-sized enterprises (SMEs) often encounter practical challenges and limitations when extracting valuable insights from the data of retrofitted or brownfield equipment. The existing literature fails to reflect the full reality and potential of data-driven analysis in current SME environments. In this paper, we provide an anonymized dataset obtained from two medium-sized companies leveraging a non-invasive and scalable data-collection procedure. The dataset comprises mainly power consumption machine data collected over a period of 7 months and 1 year from two medium-sized companies. Using this dataset, we demonstrate how machine learning (ML) techniques can enable SMEs to extract useful information even in the short term, even from a small variety of data types. We develop several ML models to address various tasks, such as power consumption forecasting, item classification, next machine state prediction, and item production count forecasting. By providing this anonymized dataset and showcasing its application through various ML use cases, our paper aims to provide practical insights for SMEs seeking to leverage ML techniques with their limited data resources. The findings contribute to a better understanding of how ML can be effectively utilized in extracting actionable insights from limited datasets, offering valuable implications for SMEs in practical settings.

## 1. Introduction

In recent years, the emergence of Industry 4.0 (I4.0) has transformed the manufacturing industry by integrating digital technologies with traditional manufacturing processes. This new era of manufacturing has brought about many benefits such as increased productivity, reduced costs, and improved quality [1]. However, the implementation of Industry 4.0 poses significant challenges, particularly for small and medium enterprises (SMEs). The challenges faced by SMEs in implementing Industry 4.0 are multifaceted and require careful consideration of the unique needs and constraints of small businesses [2].

Firstly, SMEs often have limited financial resources, and the benefits of adopting I4.0 technologies may not be as immediately evident for SMEs as they are for larger organizations. I4.0 requires significant investment in new technologies, such as the Internet of things (IoT), artificial intelligence (AI), and big data analytics, which can be costly and complex to implement. In addition, there may be a lack of awareness and understanding of Industry 4.0 among SMEs. Many SMEs may not fully comprehend the benefits of these new technologies or may not have access to the necessary information and resources to effectively implement them [3,4]. Fortunately, the growth in interest in this field and the search for more sustainable industries has resulted in funding programs and incentives from the European Union and other governments [5,6] to support SMEs in their Industry 4.0 initiatives, which can help alleviate financial burdens. In addition to this, education and training programs can be provided to increase awareness and understanding of Industry 4.0 among SMEs, enabling them to better assess the potential benefits and to justify the initial investment [7].

Secondly, the technical complexity of Industry 4.0 can be a significant challenge for SMEs, especially due to the old machinery they may have. Upgrading to the latest machines can be prohibitively expensive, and many SMEs lack the financial resources or technical expertise to make the switch. A possible solution that has been studied and adopted in recent years is retrofitting existing machinery [8]. Instead of replacing the entire machine, retrofitting allows for the integration of new elements into the existing framework. This can include adding sensors, actuators, controllers, software systems, connectivity solutions, or other advanced features that enable the machine to operate more efficiently, improve productivity, achieve higher accuracy, meet regulatory standards, or align with emerging industry trends such as the adoption of Industry 4.0 principles. Retrofitting offers a cost-effective way to leverage existing machinery investments while leveraging the benefits of modern technologies [9].

In this context, another problem that SMEs must face after retrofitting their machines is understanding why and how the data coming from their sensors can be used. The academic world has not yet been able to define a clear and definitive pathway to follow in practice, despite the fact that the growth of the Internet of things and the increased connectivity of manufacturing equipment have made it possible to collect large amounts of data from the machines and equipment used in industrial manufacturing. By now, one of the main identified benefits of analyzing data from industrial manufacturing machinery is the ability to identify trends and patterns that can help optimize production processes. By analyzing the data, it is possible to detect inefficiencies, identify opportunities for process improvements, and predict potential equipment failures [10].

In this paper, we present our work on retrofitting industrial machines in the context of discrete manufacturing environments, with a particular focus on small and medium enterprises (SMEs). Our study aims to contribute to this field in several significant ways.

Firstly, we address the need for a cost-effective and non-invasive data-acquisition phase in retrofitting processes. By leveraging power clamps and Internet of things (IoT) devices, we propose a methodology that allows for the acquisition of relevant data without disrupting the existing production status quo. This approach overcomes one of the primary challenges faced by SMEs in the discrete manufacturing sector, where implementing retrofitting solutions can be costly and disruptive. By providing a more affordable and non-intrusive data-acquisition method, we facilitate the adoption of retrofitting practices in SMEs, thereby enabling them to improve their operational efficiency without significant disruptions.

Furthermore, our work contributes to the space by releasing a dataset derived from two different companies operating in the discrete manufacturing sector. This dataset serves as a valuable resource for researchers and academics, offering them an opportunity to explore and develop a clearer pathway and methodology for retrofitting in this specific context. By sharing this dataset, we aim to encourage further research and collaboration, fostering a better understanding of the challenges and opportunities associated with retrofitting industrial machines in SMEs.

Lastly, our study provides a diverse range of machine learning (ML) and data mining use cases, showcasing the potential applications and benefits of retrofitting in SMEs. By demonstrating the practical implementation of ML techniques in this context, we aim to improve both the perception and acceptance of retrofitting practices among SMEs and the academic community. Through our use cases, we highlight how retrofitting can lead to operational improvements, cost reduction, and new opportunities for SMEs, while also presenting intriguing challenges for researchers in terms of optimizing and enhancing retrofitting methodologies.

The paper is organized as follows: In Section 1 and Section 2, we provide a comprehensive review of the relevant literature and related works about retrofitting machines. Section 3 provides an overview of the pipeline from manufacturing machines to ML results. Section 4 describes in detail the data acquisition process. In Section 5 and Section 6, we provide a thorough description of the dataset itself and some preliminary statistical analysis. Section 7 is dedicated to showcasing some of the possible machine learning applications that can be analyzed in the provided dataset. In Section 8, we present the detailed methodology for the development of our ML models. Section 9 presents the results obtained from the application of machine learning algorithms. Section 10 is dedicated to the discussion of our findings and the limitations of our study. Finally, in Section 11, we conclude the paper by summarizing the key findings, discussing their implications, and proposing future research directions.

## 2. Related Works

While there are many studies covering the topic of retrofitted machines, there are few that analyze the data coming from them. Strauß et al. [11] presented a case of brownfield digitization for heavy lifting, showcasing a general application of ML methods. To account for the scarcity of failure-labeled data in real-life scenarios, the authors used a combination of semi-supervised anomaly detection and unsupervised machine learning algorithms to enable the predictive maintenance of such machines.

In the existing literature, CNC machines are an especially covered subject when discussing legacy equipment modernization. Herwan et al. [12] proposed an alternative position for embedding sensors in CNC turning with a rotating turret, called the turret bed, and evaluated its accuracy in predicting tool flank wear through pattern classification using an artificial neural network. Thus, this study demonstrates a feasible method for retrofitting old CNC turning machines with sensors. Hesser and Markert [13] trained an artificial neural network with acceleration data to classify the tool state of a retrofitted CNC milling machine, allowing for the continuous monitoring of tool wear in service.

The process from digitization to data analysis is also described in [14], where Ralph et al. demonstrated the retrofitting of a rolling mill machine and the development of a related machine learning algorithm to predict and adapt the resulting rolling schedule of a defined metal sheet. The importance of using low-cost sensors to encourage SMEs to adopt I4.0 and to show its benefits was discussed by Lima et al. [15]. Here, the authors present a retrofit solution for a CNC milling system using an energy-measurement industrial sensor and an IoT gateway for machine connectivity and data transmission to the cloud. The embedded machine learning approach for energy prediction was described in [16], where an ANN model was used to predict the total energy consumption in job-shop environments. The authors highlighted the critical issue of input variables, demonstrated good prediction results, and suggested that the model could be used as an auxiliary tool for estimating energy consumption costs or lean energy indicators. In [17], Selvaraj and Min retrofitted an ultra-precision CNC machine with a low-cost power meter to collect power consumption data and streamed it to AWS servers in real time. Starting with these data, they developed anomaly-detection models and used machine learning to classify and cluster the different error states of the machine.

Regarding the modernization of legacy injection machines, Silva et al. [18] discussed a system using Raspberry Pi zero and Raspberry Pi 4 to collect, process and store data generated by plastic-injection machines, which can be used for machine learning algorithms and observed through dashboards created by the open-source version of Grafana. Polenta et al. [19] present a study that compares six classifiers used to predict the quality of plastic products produced by the injection molding process, using real processing parameters collected during the production of plastic road lenses (with the dataset publicly available). The results showed that the random forest classifier achieved the highest accuracy of 95.04%, confirming the suitability of ML techniques for automating quality prediction in plastic injection molding.

The literature highlights the growing interest in retrofitting legacy machines and applying machine learning techniques to analyze the data collected from them. However, several issues are identified. First, one issue that can be highlighted is the lack of utilization of easy-to-install low-cost sensors and devices, which can be noted as a barrier for SMEs to adopt Industry 4.0. Our study addresses this concern by developing prediction models for affordable sensor solutions in legacy equipment modernization scenarios, thus demonstrating the feasibility of such approaches for SMEs and making I4.0 more accessible for them.

Another major challenge highlighted in the reviewed literature is the absence of methods or models to classify and label the items produced by retrofitted brownfield equipment. This information is relevant for quality control and process optimization in various industries in the discrete manufacturing sector. To overcome this, our study compares supervised, unsupervised, and semi-supervised learning algorithms to classify produced items, laying the foundation for future research on the topic.

Third, the lack of machine-agnostic approaches in the existing literature limits the generalizability of the proposed methods. In contrast, our study adopts a machine-agnostic approach, enabling the application of our methodology and algorithms to a wide range of retrofitted brownfield equipment. In this way, our study extends the analysis to include the prediction of the next status of the machines and the forecasting of both item production and power consumption. By leveraging historical data and machine learning algorithms, we can provide accurate predictions for these key performance indicators, thus enabling better production planning, energy management, and cost optimization.

Finally, the scarcity of publicly available real-world datasets is recognized as a challenge in fostering new research on this topic. To contribute to overcoming this issue, our study releases a comprehensive real-world dataset, providing a valuable resource for researchers and practitioners to validate and compare their own approaches. By addressing these issues, our research strengthens the conclusions of the literature, facilitating the effective adoption of Industry 4.0 by SMEs and promoting advancements in data analysis and machine learning for retrofitted machinery equipment.

## 3. Pipeline Overview

The overall flow of the procedure carried out, from the retrofit machine to the ML model results, is illustrated at a high level in Figure 1. Each stage corresponds to one of the following sections in this paper. The first stage references the data acquisition enabled by retrofitting. The second refers to the data anonymization and cleaning that led to the described public dataset being released. The third represents the preliminary analysis work that allows for some understanding of the data. The last stage depicts the ML applications and their evaluation.

## 4. Data Acquisition

IoT infrastructures are complex systems that require careful design and implementation to ensure efficient and secure data collection from connected devices. For this intricate phase, we benefit from the hardware and cloud infrastructure of Zerynth (https://zerynth.com/, accessed on 25 June 2023), an Italian company working in the Industrial IoT world. In this section, we describe the IoT infrastructure we used and provide a detailed explanation of the data-collection phase.

### 4.1. Hardware

The type of device we used is a modular hardware electronic unit, called the 4ZeroBox (provided by Zerynth, Pisa, Italy), connected to a current clamp for real-time measurements of machine power consumption. A 4ZeroBox is composed of:Six analog channels that can measure: 4–20 mA sensors, 0–10 V sensors, current transformers, and resistive sensors;Two solid-state relay channels with max voltage and current equal to 36 VDC and 150 mA, respectively;RS232 and RS485 interface;CAN protocol support;Support for USB-C for PC communication and power;USB-C slot for DEBUG/updating Firmware of BG95;SMA antenna for GSM/GPRS (SX) and GPS (DX);LiPo battery support;JTAG support;Edge computation module.

The computation module is a 32 bit dual core microcontroller based on the ESP32-WROOM-32SE and allows for edge calculations that will be better described in the following section. The CPU clock frequency is treated as an option and can vary from 80 to 240 MHz. The microcontroller can also count on an embedded 16 MB of SPI flash memory, an integration of the ATECC608A crypto element allowing for ultra-secure communication, WiFi (client and AP mode supported) and Bluetooth low-energy support.

Installed in each microcontroller is a real-time operating system, the Zerynth OS, which allows for the execution of firmware written in Python and C languages. A complete guide and description of the OS can be found at https://docs.zerynth.com/latest/reference/os/, accessed on 25 June 2023. Microcontrollers allow us to perform calculations on the edge, avoiding sending data to the cloud after each measurement. Because of this, we were able to aggregate data directly at the edge; for example, we calculated statistics on measurements per minute for energy consumption, without losing the benefits of lower measurement frequency.

### 4.2. Firmware Characteristics

Creating firmware that possesses resilience and dependability within the challenging operational environment of an IoT device presents a formidable challenge. The objective is to minimize data loss in the event of potential power disruptions or lack of connectivity. To reach this goal, the firmware used in the devices presents a variety of technicalities.

The first step of the firmware for collecting and sending data is the connection to the cloud. Our devices use the MQTT protocol as well as the following functions to ensure a secure connection:Stores an ECC secp256r1 private key. The private key is inserted into the secure element by the manufacturer (microchip), exploiting its expensive and FIPS-certified infrastructure.Stores a set of device certificates. These certificates are generated by the manufacturer and are signed by its root certificate.Signs, verifies and exchanges secret keys to accelerate cryptography operations.Generates cryptographically secure random numbers, allowing for always fresh secret keys during connections.

On the cloud side, there is a copy of every certificate stored in the hardware components, and a connection can be made only if the request comes from a device owning a known certificate.

After the connection is established, the firmware will update the real-time clock to the current time and will try to keep it synchronized. The reliability of the connection is also guaranteed by automatically retrying to connect in case of network failure. Finally, data points, such as sensor readings, can be sent to the cloud in JSON format.

Regarding firmware robustness, two main features are used: a watchdog and a time series log (TSLog). In real-world products, it is imperative to include a watchdog mechanism that automatically resets the board if the firmware becomes unresponsive. This crucial feature ensures the stability of the system by initiating a board reset after a specific time frame if the firmware fails to provide periodic signals to the watchdog, indicating proper functionality.

TSLog offers a solution for separating data acquisition and transmission in time series applications. In this use case, multiple sensors generate data points that are not immediately sent over the network. Instead, they are stored in the log system, allowing a separate thread to access the log in read-only mode and to retrieve the data points for transmission when a network connection becomes available. The architecture of TSLog ensures its resilience and ability to withstand challenges commonly faced in IoT scenarios, such as network unavailability, system resets, and power loss.

TSLog stores fixed-length records in containers known as buckets, residing in non-volatile memory. Each record is assigned a sequentially increasing sequence number. As a bucket reaches its capacity, a new bucket is created to accommodate incoming records. At regular intervals, the buckets are committed, meaning a snapshot of the current log status is taken and saved in non-volatile memory. Upon successful snapshotting, all records up to the current sequence number are securely stored and ready for retrieval. To facilitate data extraction, TSLog employs readers, specialized objects capable of reading records from the buckets. Each reader maintains its own cursor, representing the sequence number of the last successfully read record. Readers also have the ability to commit their state, permanently saving the value of their cursor. Due to these characteristics, TSLog proves to be a robust solution in cases of power loss or network unavailability. When the system resumes operation, all readers automatically restart from the last committed cursor, ensuring data continuity. Moreover, the log system automatically cleans up old data by deleting buckets that have been completely consumed by all readers.

### 4.3. Cloud Infrastructure

In this paper, we leverage the Zerynth cloud platform that encompasses various key characteristics to facilitate the deployment of scalable and secure IoT solutions. The architecture of the cloud platform revolves around two main components: device management and data storage, complemented by the availability of REST APIs for seamless integration and extensibility.

Device management is a software platform for handling hardware devices. It allows one to control the devices’ life cycle with remote updates. In addition, as previously mentioned, each physical device is linked with one virtual device via a hardware components certificate. Thus, with the adoption of strict security measures, devices can be easily added to the device management while ensuring data integrity and confidentiality.

Data storage is a dedicated cloud service optimized for storing time series data. This service provides efficient storage and retrieval mechanisms for both raw and aggregated data and offers the flexibility to export data in various formats such as CSV and JSON.

## 5. Dataset Description

Our dataset was collected using the infrastructure described in the prior section, from two companies, “A” and “B”. The rows were produced at a 1 min frequency. Listings 1 and 2 show the final attributes of the dataset. The first two listings refer to Company A and B’s data attributes. A small separate subset of Company A’s data was labelled manually by operators to include what item was being produced. The subset consists of 14,491 rows at a 5 min frequency spanning 20 days and is described in the Listing 3.

**Listing 1.** Company A Data Attributes.
{

  ts: (string) a timestamp in the format YYYY-MM-DD HH:mm:ss+TZ,

  asset: (string) an identifier for the machine used;

  items: (int) the number of items produced in the time span;

  status: (int) the machine state where 0 is idle, 1 is manual production mode;

  2 is automatic production mode and 3 is an alarm or interrupted state;

  power_avg: (int) the average power consumed, in kilo-watts;

  cycle_time: (int) the total time, in seconds, to produce the items.

}


**Listing 2.** Company B Data Attributes.
{

  ts: (string) a timestamp in the format YYYY-MM-DD HH:mm:ss+TZ;

  asset: (string) an identifier for the machine used;

  status: (string) the machine state

  (Alarm, Standby, MachineOn, Production, Loading, Tooling);

  alarm_time: (int) the time spent in the alarm state;

  loading_time: (int) the time spent preparing the machine for production;

  tooling_time: (int) the time spent preparing machine tools;

  maintenance_time: (int) the time spent performing machine maintenance;

  support_time: (int) the time spent performing machine repair;

  power_avg: (int) the average power consumed, in kilo-watts;

  power_max: (int) the highest power consumption value in the 1-minute period;

  power_min: (int) the lowest power consumption value in the 1-minute period.

}


**Listing 3.** Company A Subset with Product Labels.
{

  ts: (string) a timestamp in the format YYYY-MM-DD HH:mm:ss+TZ;

  asset: (string) an identifier for the machine used;

  items: (int) the number of items produced in the time span;

  status: (int) the machine state where 0 is idle, 1 is manual production mode,

  2 is automatic production mode and 3 is an alarm or interrupted state;

  power_avg: (int) the average power consumed, in kilo-watts;

  cycle_time: (int) the total time, in seconds, to produce the items;

  product: (int) a product identifier ranging from 0 to 13.

}


For company A, there is a total of 1,521,065 rows over a 7-month period using 9 machines. While for company B, there is a total of 1,568,736 rows over a 1-year period using 18 machines. The dataset also contains measurement errors that we decided to keep in order to allow users to adopt different preprocessing techniques. In our work, we drop the rows with outliers in the power-consumption-related columns and replace values bigger than 60 in the time-related columns, since they represent seconds spent in the machine in different machine states.

Despite the increasing interest in machine learning for industry 4.0 applications, there are not many public data options for time-series discrete manufacturing. Hence, we saw the need to make our dataset public as a contribution. It is available at https://github.com/HumanCenteredTechnology/SME-Manufacturing-Dataset, accessed on 25 June 2023.

## 6. Preliminary Analysis

In order to (1) gain a basic understanding of the data distribution of values and (2) to examine the possible limits of what can be carried out with simple statistical techniques (i.e, without ML), we also performed a preliminary analysis of our dataset.

We initially calculated some descriptive statistics (such as averages, standard deviations and minimum, maximum and percentile observed values) in order to gain a basic understanding of the data. Table 1, Table 2 and Table 3 show the results of this for each company and feature.

Toward our second goal for this step, we computed the pairwise correlation of the features for each company. The plot for company A is shown in Figure 2. From this figure, we can see that there is a limited correlation between any two given features, thus limiting what one can achieve purely with simple statistical inferences. On the other hand, ML approaches can explore deeper relationships with the data and a target by finding more complex representations of the features. The plot for company B was quite similar; thus, we chose to omit that figure.

## 7. Machine Learning Applications

As previously said, the I4.0 paradigm has transformed the manufacturing industry by leveraging automation, interconnectivity, and data exchange. In the context of data analysis, artificial intelligence and machine learning have recently shown their power and the impact they can have in a variety of human activities, such as text generation [20], image synthesis [21], biology [22], human behaviors [23], etc.

For this reason, it should not be surprising that they can also be key technologies driving the fourth industrial revolution. Existing examples of machine learning applications in Industry 4.0 involve predictive maintenance [24], demand forecasting [25], and scheduling activities [26]. The authors in [27,28] provide a complete list of interesting applications as well as challenges and opportunities for machine learning in I4.0.

In this section, we describe some possible applications of ML and AI in the retrofitting scenario that, despite what we said above about success, remains still little explored. We also tested some algorithms trained on the previously mentioned dataset. As described in Section 5, the main available features in the dataset are the number of items produced, machine status and power consumption. Combined with the available supervised data about the type of items being produced, we will use AI and ML algorithms for three different applications: next status prediction, item classification, and item production count and power consumption forecasting.

### 7.1. Next Status Prediction

The first possible use of machine learning we analyze is machine status prediction. As previously said, the available data offer us three different machines statuses: working, idle, and alarm.

The most straightforward use case of this ML application is the prediction of alarms. Clearly, a manufacturing company can derive several benefits from early predictions of alarms, such as predictive maintenance and worker safety. Further analyses can also include the adoption of explainable AI methodologies [29] to try to understand the patterns in the data that lead to an alarm status.

Another possible advantage this use case can provide is early knowledge of the duration of working and idle phases. This knowledge can be very useful for the optimization of production processes, which is one of the main challenges for discrete manufacturing companies. Several activities can improve with a full understanding of the production stage, such as scheduling activities or supporting managerial decision making for the production phase.

### 7.2. Items Classification

The second application we explore is item classification. Our data provide only information about the number of actions each machine performs in every time step (e.g., the number of presses for injection molding machines in one minute), without any information about the type of item that is being produced. In addition, manual labeling of the items being produced is a tedious and intense task. For this reason, we decided to test AI and ML algorithms that can leverage patterns in the data to extract the types of items that are being produced. This knowledge can be used for a better understanding of the real cost of items as well as for the production time, in order to improve the scheduling process and to optimize cost or revenues.

The availability of a portion of labels for a subset of machines allows us to compare three different machine learning paradigms: unsupervised, semi-supervised and supervised. The unsupervised paradigm allows us to analyze and cluster unlabeled data. Unsupervised algorithms usually leverage some definition of distance between data points and try to find a (predefined or not) number of clusters by minimizing distances between points in the same cluster and maximizing them between points in different ones. Another option is to include another step before the clustering phase, often used for non-tabular data points, in which data points are mapped into vectorial representations [30], possibly followed by a feature reduction algorithm. It is the most challenging but also the most likely to be impactful, since it does not require any labeling phase, which can be cumbersome and costly both for workers and companies.

Unlike unsupervised techniques, supervised AI and ML classification algorithms require complete knowledge of data labels and aim at minimizing the number of classification errors. Technically speaking, this paradigm is the easiest, since it leverages a much bigger amount of information with respect to the others. This paradigm is also the most studied in the literature, but the challenging data-acquisition process makes its impact smaller than expected in real-world scenarios.

Semi-supervised, or weakly supervised, classification falls in between the previously described paradigms. It combines the knowledge of a small portion of labels with a large amount of unlabeled data. In the analyzed context, this may seem the most suitable, since it represents a compromise between the tedious task of collecting labels and the difficulty of the use case. Many different methods have been developed in the literature, and they range from probabilistic generative mixture models [31] to graph-based methods, where similar data points are connected and the labels are used for classical algorithms [32] and deep-learning-based methods [33].

### 7.3. Item Production and Power Consumption Forecast

The last application we analyzed is the forecasting of both power consumption and item production.

Power consumption forecasting is important because it enables manufacturers to identify areas where energy usage can be reduced, which can lead to significant cost savings. By forecasting power consumption, manufacturers can also ensure that they have enough energy to meet production demands. This is particularly important in the context of Industry 4.0, where manufacturers increasingly rely on real-time data to make informed decisions [34].

Item production forecasting plays a pivotal role in driving efficiency, optimizing resource allocation, and enhancing overall productivity. As manufacturing processes become increasingly interconnected and automated, the ability to accurately predict item production has become essential for seamless operations. This proactive approach minimizes inventory costs, reduces wastage, and ensures timely delivery, thereby maximizing customer satisfaction. Moreover, accurate production forecasting empowers companies to make informed decisions regarding capacity planning, resource utilization, and workforce management, leading to improved operational agility and competitiveness in the rapidly evolving Industry 4.0 landscape [35].

## 8. Methods

In this section, we describe the procedure we adopt for developing ML models to address the aforementioned applications. For all of them, we split the dataset into two splits. The first one, composed of 80% of the data, is used for model training and validation (for supervised tasks). The remaining part is used as a test set to assess model performances. Model selection and evaluation are performed by a grid search of model hyperparameters and k-fold cross-validation [36], with k=5. The model-selection phase mainly focused on two types of models: random forest [37] and LSTM [38]. We made this decision given the proven strength and past successes of these models [39,40,41]. That being said, we recognize the modest depth of our model selection phase and grid search, thus we acknowledge that exploring these aspects extensively falls beyond the scope of our study. Our primary objective was to showcase and establish the profound impact of machine learning by developing models that are deemed sufficiently effective by using almost only power-consumption data. Table 4 shows the hyperparameters used for each model.

In the following subsections, we elaborate on the technical details for each of the machine learning paradigms we faced: supervised regression for item count and power-consumption forecasting, supervised classification for the next state prediction and item classification, and unsupervised and semi-supervised learning tasks.

### 8.1. Supervised Regression Tasks

Predicting a continuous variable based on machine data and prior patterns is a problem that can be shaped into many forms depending on the forecasting horizon and available data. The further ahead one aims, the more difficult the problem, but the closer to the current time, the less useful the information. We used mean absolute error (MAE) to validate and assess our models.

Regarding power consumption, focusing more on usefulness, we chose to predict the total power consumption for the next day of manufacturing. We used company B for this model due to its larger variety of machines and therefore greater variability in power consumption. We opted for an LSTM model to exploit the temporal structure of our data.

Instead, in the item count scenario, we decided to illustrate the problem in its simplest form, predicting the number of items produced in a period based on the machine data of that period, as a basis for expanding into its more complicated forms. For this, we trained a random forest regressor on company A’s data using all the features available.

### 8.2. Supervised Classification Tasks

Supervised classification tasks include next-state prediction and item classification by leveraging the portion of the data that includes item labels. To evaluate the models, we used accuracy for both cases.

The problem of identifying items being manufactured based on machine sensor data faces the challenge of limited data availability in practice. In our described dataset, only 14,492 rows have labels on what items were being produced. Moreover, as we can see in Figure 3, the labels are not evenly distributed across the possible classes. For this reason, we trained a random forest classifier both on the original unbalanced dataset and on a balanced one obtained by oversampling [42] the original dataset.

The machine alarm state represents an unexpected interruption. Toward exemplifying the case of predicting the next alarm state, we trained a random forest classifier to predict whether or not there will be an interruption within the next 5 min given the current minute of data. This choice allows us to have a balanced dataset without compromising its real-world plausibility. For company A, some machines rarely enter an alarm state; thus, we trained and evaluated using only one machine that did so relatively frequently. For company B, we trained and evaluated a model using each machine separately and calculated the average F1 score.

### 8.3. Unsupervised and Semi-Supervised Tasks

As previously said, manually labeling data is a tedious, expensive and error-prone operation. For this reason, aiming at having a more realistic impact in real-world scenarios, we tested two different machine learning paradigms for the item classification task: unsupervised and semi-supervised learning. For the second one, we focused only on semi-supervised clustering. In particular, we tested the constrained k-means clustering [43]. Both unsupervised and semi-supervised clustering methods were evaluated by calculating the adjusted rand score [44] between the true labels and the predicted clusters.

The unsupervised model was based on simple k-means clustering. Despite knowing the ideal number of clusters, i.e., the number of possible items, we also performed the clustering by selecting the ideal k, according to Bayesian information criterion (BIC) [45] and Akaike information criterion (AIC) [46]. Then, we analyzed the item distribution in each cluster to check whether our partition was able to discriminate item types or not.

## 9. Results

The two regression tasks turned out to be the most complex ones. The best model for power consumption forecasting yielded a mean absolute percentage error of 16%, while the item production count forecasting model gave an $R^{2}$ score, also known as the coefficient of determination, of 0.902.

The next alarm-state prediction, instead, gave better results. The model resulted in an average F1 score of 84.7% on company A’s machine 0 and an average of 88.6% across all of company B’s machines.

Given the unbalanced nature of the available labels, for the supervised item classification, we trained a random forest model on both the original dataset as well as on an oversampled one. Oversampling was carried out in such a way that the minority classes represented at least 20% of the majority class. In both cases, the accuracy score was 74%, and we did not find any major difference between the techniques. We also computed model performances in top-three classification, i.e., we checked whether the correct label was within the three most probable predicted classes. This score may be useful for understanding the feasibility of an application of the ML model that can support and facilitate data annotation, offering fewer alternatives to a human operator. The best model obtained 97% of accuracy.

The unsupervised task was carried out with k-means. Based on the knowledge about the number of different types of items, we decided to make an initial attempt by fixing k=14, trying to see if each type of item can be grouped into a single cluster. This attempt yielded a sum of squared distances from cluster centroids of 5773 and an adjusted rand score with respect to the actual labels of 0.39. This score ranges between −0.5 and 1, with values being close to 0 for random labeling and 1 for a perfect match. For reference, generally, a score between 0.25 and 0.5 is considered to indicate moderate clustering performance. The second analysis we performed was checking the ability of k-means to discriminate between types of items with the ideal number of clusters. To find it, we computed BIC and AIC scores for values of *k* up to 130. The optimal value with respect to these scores was 34, and the sum of the squared distances in this case was 1608. To analyze clustering discrimination ability, we used the frequencies of labels within a cluster as posterior probabilities assigned to a data point belonging to that cluster. In doing so, we computed both accuracy and top-three accuracy and obtained 58% and 92%, respectively.

Finally, the semi-supervised clustering did not give the expected improvements compared to unsupervised clustering. For constrained k-means, we obtained an adjusted rand score of 0.35, even smaller than the unsupervised k-means. Understanding this behavior and trying different semi-supervised clustering and learning techniques is part of our planned future work, given the possible impact of the semi-supervised paradigm in real-world scenarios.

## 10. Discussion and Limitations

The total power consumption forecasting use case is a challenging one because its usefulness is tied to the length of the prediction horizon. Despite this issue, we saw moderate performance from the respective model with a mean absolute percentage error of 16%. We believe this helps to highlight the potential of retrofitting’s usefulness. However, in some environments where the daily production schedule is consistent, the most likely power consumption total is easier to predict but is not useful, whereas predicting an anomalous value ahead of time is more relevant but is even more challenging.

The aforementioned issues also extend to the case of item production count forecasting, where the larger the forecast horizon the more challenging it is to predict. For this reason, we focused on exploring the simplest case of predicting the production count given only the machine data. The result of this was an $R^{2}$ of 0.902, suggesting a strong predictive relationship. We believe this bodes well for other expanded use cases.

The item classification solution could enable many additional routes for impactful insights into company data. For instance, having individual labels on item production as machines produce them would allow for analytics on a deeper item-by-item scale than what is otherwise possible. Manual labeling for this is tedious and time-consuming and is prone to human error considering the high frequency at which item production occurs. This also means that the problem is challenging for machine learning to solve due to the lack of data in sufficient quantity and quality.

That being said, the threshold for the usefulness of such a model is lower than those already discussed. Because even though attaining high accuracy in predicting the exact item is difficult, if the correct answer is within the top three or five predictions, it can be used for aiding the labeling process. Operators can select the correct answer from them on a screen. This, in turn, provides a stepping stone for better solving the problem in the future while allowing the company to benefit in the present.

The unsupervised result indicates moderate performance, which is also a promising sign for solutions that address the lack of individual labels as overall machine data become more available. On the contrary, the semi-supervised approach did not produce the expected improvements over the unsupervised one. These unexpected results should be more carefully analyzed in future works, and more advanced semi-supervised techniques should be used.

Moreover, from the analysis of these different approaches to the problem we noted, the power consumption profile is potentially a strong indicator of the item being produced. Figure 4 shows the average power consumption over a 13-day period from the company A subset with product labels. It shows that the distinction between the three products produced can be seen from only the power consumption values.

Further leveraging of this relationship is of significant importance. If a solution to the item classification problem can be carried out using only this one feature, it reduces the required components for retrofitting the machines. This in turn allows companies to benefit from the analyses with less upfront cost and effort.

Lastly, the results of the alarm state prediction, an average F1 score of 84.7% on company A’s machine 0 and an average of 88.6% across all of company B’s machines were positive considering the potential impact of even a 5-minute horizon period. However, the more challenging cases of machines with infrequent alarm interruptions are likely to need more advanced and tailored solutions.

Overall, we are aware that the reported results can be improved through a more detailed and wider model-selection phase. However, this goes beyond the scope of this work, since our goal is to show possible data analyses that can be realized using a cheap and potentially large-scale data-acquisition system that is suitable for SMEs. We believe that the proposed ML applications exemplify the potential for usefulness considering: (1) the limited time for collection, (2) the leverage of almost only machine power consumption, (3) the higher granularity of the data due to the network flow and size limitations that come with storing higher frequency data, and (4) the fact that value could still be extracted from approximate predictions.

## 11. Conclusions and Future Work

Given the practical challenges and limitations of extracting useful insights from the data of retrofitted or brownfield equipment, much of the existing literature does not reflect what is possible in current SME environments.

Toward addressing the data availability issue, we present an anonymized dataset of discrete manufacturing machine data from two medium-sized companies over 7-month and 1-year collection periods. Using these data, we showcase how ML can help companies extract useful information even in the short term as they work toward building their own historical datasets. We put forward several ML model use cases, power consumption forecasting, item classification, next machine state prediction and item production count forecasting, the results of which exemplify the potential impact despite the challenging circumstances.

Future work includes exploring more recent machine learning approaches to develop new applications or to simplify the present ones. An interesting paradigm can be continual learning, which can help models reach a useful state in a shorter time, as well as allow for a more flexible model update phase. Another interesting ML paradigm is explainable AI, which can be used to provide a clearer description of model predictions and to extract human-understandable knowledge from ML models. Another possible improvement over the current approach is to move AI and ML models from cloud to edge devices. This can overcome the limitations leading to coarse data granularity.

Lastly, a major direction of planned future work is aimed at inferring the machine state and item cycles based solely on the power consumption profile. Such a feature would minimize the components that need to be added to retrofitted machines while still helping companies gain the benefits of item classification and analyses.

## Figures and Tables

**Figure 1 sensors-23-06078-f001:**
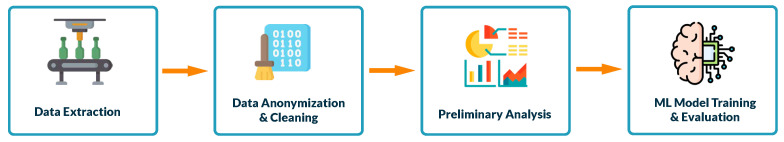
Flow chart of procedure.

**Figure 2 sensors-23-06078-f002:**
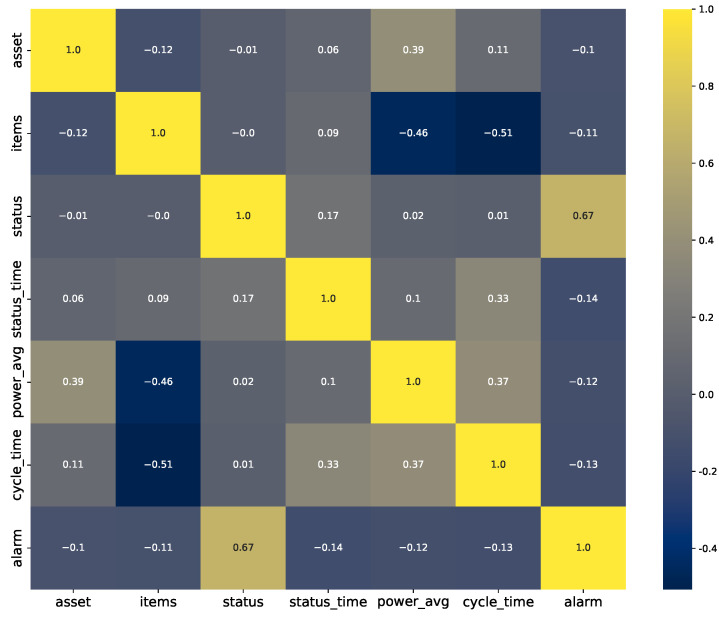
Correlation plot for Company A.

**Figure 3 sensors-23-06078-f003:**
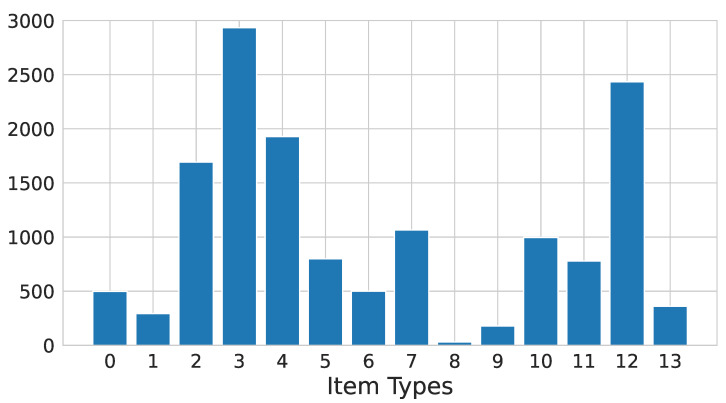
Distribution of item labels.

**Figure 4 sensors-23-06078-f004:**
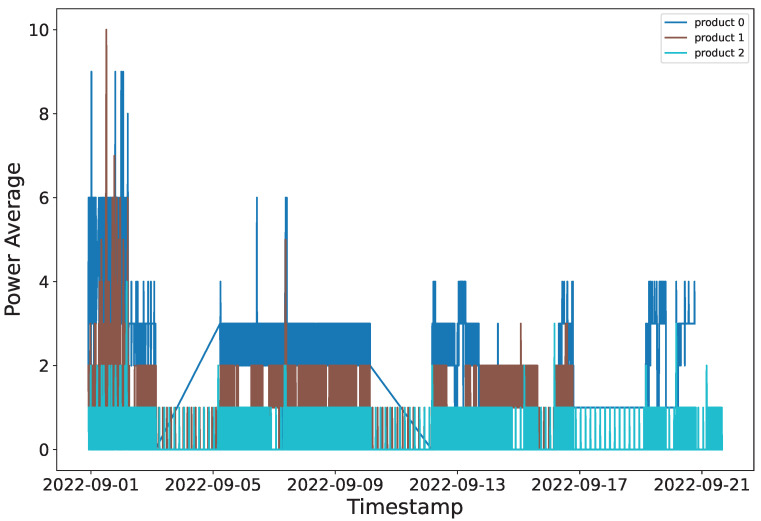
Power consumption over 13 days of producing three item classes.

**Table 1 sensors-23-06078-t001:** Company A data statistics.

	Asset	Items	Status	Status_Time	Power_Avg (kW)	Cycle_Time (s)	Alarm
mean	3.882	2.152	1.609	56.427	39.891	23.302	0.075
std	2.601	3.103	0.623	12.623	19,159.073	27.986	0.264
min	0.000	0.000	1.000	0.000	0.000	0.000	0.000
25%	2.000	0.000	1.000	60.000	0.000	0.000	0.000
50%	4.000	0.000	2.000	60.000	0.000	0.000	0.000
75%	6.000	4.000	2.000	60.000	3.000	60.000	0.000
max	8.000	547.000	3.000	60.000	16,777,216.000	60.000	1.000

**Table 2 sensors-23-06078-t002:** Company B data statistics.

	Asset	Alarm_Time (s)	Loading_Time (s)	Tooling_Time (s)	Maintenance_Time (s)
mean	7.860	1.918	2.732	0.834	0.694
std	4.703	10.289	11.962	6.989	6.370
min	0.000	0.000	0.000	0.000	0.000
25%	4.000	0.000	0.000	0.000	0.000
50%	8.000	0.000	0.000	0.000	0.000
75%	12.000	0.000	0.000	0.000	0.000
max	17.000	62.000	61.000	60.000	62.000

**Table 3 sensors-23-06078-t003:** Company B data statistics continued.

	Support_Time (s)	Power_Avg (kW)	Power_Min (kW)	Power_Max (kW)
mean	0.191	46.417	29.913	74.176
std	3.374	72.841	48.924	108.651
min	0.000	0.017	0.000	0.017
25%	0.000	3.504	2.714	5.374
50%	0.000	7.904	5.038	14.778
75%	0.000	75.933	44.335	109.232
max	60.000	31,241.600	28,171.600	32,821.800

**Table 4 sensors-23-06078-t004:** Hyperparameters for each model.

Use Case	Model Type	Hyperparameter	Value
Power consumption forecasting	LSTM	hidden size	32
		learning rate	0.01
		dropout rate	0.02
Item classification	Random Forest	n_estimators	1000
Next state prediction	Random Forest	n_estimators	100
Item count forecasting	Random Forest	n_estimators	10

## Data Availability

The datasets presented and utilized in this study are available at https://github.com/HumanCenteredTechnology/SME-Manufacturing-Dataset, accessed on 25 June 2023.

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
