# Peer review of "Data-Driven Insights through Industrial Retrofitting: An Anonymized Dataset with Machine Learning Use Cases"

_sensors, 2023, doi:10.3390/s23136078_

Round 1

Reviewer 1 Report

The article “Data-based Analytics through Industrial Modernization: an anonymized dataset with examples of use in machine learning” is devoted to machine learning (ML). ML can be used to predict energy consumption, classify products, predict the condition of a test machine and predict the amount of production time. Management support for two small and medium-sized enterprises. Using machine learning techniques, software developers have developed predictive models that can accurately predict and identify useful patterns, providing proactive energy management strategies in industrial environments and facilitating automated quality control and inventory management processes. Indeed, the best energy consumption forecasting model gave an average absolute percentage error of 16%, while the atem production quantity forecasting model gave an R2 estimate of 0.902. Further directions for the development of this development in the field of ML are proposed.

The article can be published in present form.

The article is written without gross grammatical errors. Detailed sentences were used. The authors showed a good level of English Language.

Author Response

Thank you for your review and feedback on our article. We appreciate your positive comments on the quality of the English language used in the article. We made some changes following suggestions given by other reviewers. In particular, we mainly modified Abstract, Introduction, and Related Works to clarify context and contribution of this work, and highlight the shortcomings of the current state of the literature. 

Any comment about the modified sections is still really appreciated.

Reviewer 2 Report

Dear authors, I have already checked your manuscript entitled "Data-Driven Insights through Industrial Retrofitting: An Anonymized Dataset with Machine Learning Use Cases". After reviewing the manuscript. I found that the main contributions and benefits of your research. The recommendations for this decision are as follows:

1. In line 103, “…., we aim to develop predictive models….” should be changed into “…., it aims to develop predictive models….”.

2. In line 109, “In [12], Strauß et al. presented…” should be changed into “In what? Strauß et al. [12] presented…….”.

3. In line 115, “Herwan et al. [13] proposes…” should be changed into Herwan et al. [13] propose…”.

4. In line 163, “…we used and describe in detail the data collection phase.” should be changed into “…we used and represent in detail the data collection phase.”.

5. The author needs to explain in detail in the "Introduction" what is the effects of An Anonymized Dataset with Machine Learning. Moreover, the significance and progress of your research needs to be explained in "introduction" section. Some references should be cited as follows: (a) Synergistic effects of dodecane-castor oil acid mixture on the flotation responses of low-rank coal: A combined simulation and experimental study. International Journal of Mining Science and Technology 2023; 33:649-658. (b) Investigation of collector mixtures on the flotation dynamics of low-rank coal[J]. Fuel, 2022,327: 125171.

6. The “Conclusions” and “Introduction” sections should be refined and shorten.

7. The quality of the figures should be improved.

8. Authors should carefully check the format of references and citations.

no.

Author Response

Thank you for your time and your review and feedback on our article. We modified the sentences as you suggested in the first 4 points of your review. We refined and shortened the introduction, trying to make it more clear and smooth. We reviewed the conclusion, as it also includes the “future work” content, we could not see a way to shorten it further. We improved the clarity of figures by changing to a better quality format and improving the clarity of axes labels. We unified the format of the references and citations. Regarding point 5 of your review, we examined the suggested papers to cite and as they are focused primarily on molecular dynamics we could not see how to use them as examples that support the need for our research. If you could please provide more details about how and where these works could improve the robustness of our work, we will be happy to enrich our bibliography.

Reviewer 3 Report

The article contains information technical and innovative. The problem addressed is current and has technical relevance, which makes it significant. The paper is well-organized and convincing. The experimental methodology is described comprehensively. The results justify interpretations and conclusions. 

My recommendations are:

* The abstract can be rewritten to be more meaningful and should clarify what is proposed (the technical contribution) and how the proposed approach is validated.

* Literature review techniques must be strengthened by including the issues in the current system and how the author proposes to overcome them.

* In the references in the Introduction section, the authors cite some works. However, they have not indicated the advantage or disadvantage and their relations to this paper. It's a little confusing.

* The author should depict the flow graph to illustrate the need for the proposed approach.

* Quality of Figures is so important too. Please provide some high-resolution figures. The comparison of different methods using clear graphs should be explained.

* Comparisons with recent studies and methods would be appreciated. 

* The paper does not provide significant experimental details needed to correctly assess its contribution: What is the validation procedure?

Author Response

Thank you for your time and your review and feedback on our article. We made changes to the abstract (beginning on lines 4 and 8) to improve the clarity. We also extended the Related Works section by highlighting current SotA limitations and the lack of possibility to compare what we did with other works. This was also in response to the comment on comparisons with recent studies and methods. We refined and shortened the introduction, trying to make it more clear and smooth, focusing on current existing problems and the contribution of our work. We improved the clarity of figures by changing to a better quality format and improving the clarity of axes labels. We also added a small section summarizing the pipeline of work done, in response to the request for a flow graph. Finally, we tried to better describe the methodology to develop and validate our Machine Learning models in the “Method” section.

Round 2

Reviewer 2 Report

The paper has been well modified and can be accepted.

no.

Reviewer 3 Report

The authors made all the required changes. I am in favor of its publication.